# African American Women with Cardiometabolic Complications of Pregnancy Have Decreased Serum Abundance of Specialized Pro-Resolving Lipid Mediators and Endocannabinoids

**DOI:** 10.3390/nu15010140

**Published:** 2022-12-28

**Authors:** Kristal M. Maner-Smith, Erin Ferranti, Anne Dunlop, Elizabeth Corwin, Eric A. Ortlund

**Affiliations:** 1Emory School of Medicine, Emory University, Atlanta, GA 30322, USA; 2Nell Hodgson Woodruff School of Nursing, Emory University, Atlanta, GA 30322, USA; 3School of Nursing, Columbia University, Atlanta, NY 10032, USA; 4Department of Biochemistry, Emory University, Atlanta, GA 30322, USA

**Keywords:** docosahexaenoic acid (DHA), polyunsaturated fatty acids (PUFA), oxylipins, endocannabinoids, specialized pro-resolving mediators (SPM), gestational diabetes (GDM), gestational hypertension, preeclampsia, nutrition, lipidomics

## Abstract

African American (AA) women experience higher rates of maternal morbidity and mortality compared to US women of other racial/ ethnic groups. Cardiometabolic complications of pregnancy (including gestational diabetes, gestational hypertension, and preeclampsia) are leading contributors to maternal morbidity and mortality. Marked changes in circulating lipids are known to accompany cardiometabolic complications of pregnancy. Serum concentrations of docosahexaenoic acid (DHA) have been shown to be inversely correlated with risk for preeclampsia. DHA is a biosynthetic precursor of a class of specialized pro-resolving mediators (SPMs), resolvins, that have anti-inflammatory properties and are also associated with hypertensive disorders of pregnancy. We employed targeted lipidomics to characterize the distribution of DHA-containing phospholipids and SPMs in maternal serum collected in early and late pregnancy (8–14 weeks and 24–30 weeks gestation, respectively) to identify key lipids that are dysregulated during pregnancy in AA women who develop cardiometabolic complications. We identified a lipid signature in early pregnancy serum samples of AA women that is predictive of cardiometabolic complications of pregnancy with 74% accuracy. These are Resolvin D1, Resolvin E1, 2-AG, PGE2-glyerol ester, and 36:6 PC. These findings suggest that there are blood-based markers detectable in early pregnancy that can potentially identify persons at risk and tailor clinical interventions.

## 1. Introduction

Docosahexaenoic acid (DHA) is a long chain polyunsaturated fatty acid (LCPUFA) that belongs to the omega-3 family of fatty acids, implicated in the proper development and function of the central nervous system [1,2,3]. DHA, in particular, has been shown to have cardioprotective benefits [4,5,6] as it is a regulator of K^+^ channels and leads to vasodilation in pulmonary hypertension [7,8,9]. DHA supplementation has also been found to decrease gene expression of inflammatory cytokines interleukin-1 (IL-1) and tumor necrosis factor α (TNFα) and increase gene expression of peroxisomal proliferator-activated receptor γ (PPARγ) in women with gestational diabetes. For these reasons the American College of Obstetricians and Gynecologists (ACOG) recommends that women eat two weekly servings of 8–12 ounces of fish or shellfish prior to, during pregnancy, and while breastfeeding [10]; the International Society for the Study of Fatty Acids and Lipids recommends that pregnant women consume at least 200 mg DHA/day [11]. Despite these recommendations, DHA intake in pregnant women remains low and may contribute, in part, to pregnancy complications including preterm birth and the conditions related to preterm birth such as hypertensive disorders of pregnancy [12,13]. DHA lowers TNFα and markers of oxidative stress, leading to decreased systolic and diastolic pressures in rats [14]. Interestingly, maternal DHA levels in African American (AA) women have been reported to be lower in comparison with pregnant women of other racial or ethnic backgrounds [15], which is likely reflective of racial and socioeconomic disparities in preconception and prenatal diet quality and supplementation [16,17,18].

Gestational diabetes, gestational hypertension and preeclampsia are common cardiometabolic complications of pregnancy [19,20,21] and are associated with adverse pregnancy and perinatal outcomes as well as future cardiometabolic disease for both the woman and her offspring [20,22,23]. African American women are disproportionately burdened with both hypertensive disorders of pregnancy (HDP) [24] and experience greater morbidity with gestational diabetes [25]. Additionally, AA women are at significantly increased risk of preterm delivery compared to women of other races and ethnicities [24,26,27,28,29], with recent national health statistics showing that AA women have 50% greater risk of preterm delivery compared to white women [30]. This health disparity is only marginally reduced when controlling for factors related to socioeconomic status, such as level of education and access to adequate medical care [31,32].

Biomarker investigations that are associated with preterm delivery have yielded different biomarker patterns for African American compared with white women [22,24,33,34]. An ingenuity pathway analysis (IPA) of plasma biomarkers of preterm delivery indicated that African American women had significant increases in biomarkers related to cell cycle function and inflammation, including interleukin-1b (IL-1*b*), interleukin-6 receptor (IL6R), interleukin-8 (IL8), macrophage inflammatory protein-1a (MIP1*a*), tissue inhibitor of metalloproteinases -1 (TIMP1), TNF*a,* and tumor necrosis factor receptor −1 (TNFR-I), whereas white women had significant increases in biomarkers associated with hematological factors and inflammation, including interleukin-1 receptor antagonist (IL1RA), interleukin-8 (IL8), TNFR-I, and vascular endothelial growth factor (VEGF). Combined, these findings suggest that the predominant pathways that result in preterm delivery and cardiometabolic complications of pregnancy vary significantly between African American and white women [35]. Interestingly, a subset of DHA derived specialized pro-resolving mediators (SPMs) has been found to significantly decrease the abundance of pro-inflammatory cytokines, IL6, TNFα, and others [36,37].

In this study, we used targeted lipidomics to selectively target the subset of DHA-containing phospholipids and SPMs in the serum of pregnant African American women with and without cardiometabolic complications of pregnancy to determine if key DHA-derived lipids were altered over the course of pregnancy. Secondarily, we used a Receiver operating characteristic (ROC) curve to determine if the serum abundance of the identified lipids is predictive of cardiometabolic complications of pregnancy in early pregnancy.

## 2. Materials and Methods

### 2.1. Study Design and Population

This nested case–control study was undertaken on a subset of pregnant women who were invited to participate in an on-going study cohort in the Emory University African American Vaginal, Oral, and Gut Microbiome in Pregnancy Cohort Study, which is described in detail elsewhere [31]. Briefly, self-identified AA women aged 18–40 years pregnant with a singleton pregnancy presented at 8–14 weeks gestation to clinics affiliated with Grady Memorial Hospital, a publicly-funded hospital, or Emory Midtown Hospital, a private hospital, were invited to participate in this on-going cohort study. Exclusion criteria for study entry include the presence of any chronic medical condition, including chronic hypertension. Those who meet criteria and provide informed consent undergo data collection at the time of enrollment, Visit 1 (8–14 weeks gestation) and again at Visit 2 (24–30 weeks gestation). The women whose data were included in this nested case-control study were chosen from those who enrolled between June 2014, and August 2015, who had a venous blood sample collected at 8–14 weeks gestation and again at 24–30 weeks gestation, both yielding serum samples suitable for lipidomics analysis and processing. The women in this set selected as cases were those who had a diagnosis of a cardiometabolic complication of pregnancy including gestational diabetes (*n* = 7), gestational hypertension (*n* = 9), or preeclampsia (*n* = 11), for a total *n* = 27, while controls were those whose pregnancy resulted in a full-term birth (between 39–0/7 and 40–6/7 weeks of gestation) without cardiometabolic complications of pregnancy (*n* = 27).

### 2.2. Data Collection

Data collection occurred at the enrollment Visit 1 (between 8–14 weeks gestation) and Visit 2 (24–30 weeks gestation) via medical record abstraction completed post-delivery to capture the birth outcomes and complications that occurred over the course of the pregnancy. Measures collected as part of the larger cohort study that are relevant to this study include the following:

Socio-demographic Survey based on maternal self-report and prenatal administrative record review was used to ascertain maternal age, years of education, marital status, health insurance status, and household income in relation to the federal poverty level. Relevant patient demographic information is de-identified and provided in Table 1.

*Medical Chart Abstraction* was completed by the research team using a standardized chart abstraction tool to ascertain for the following pre- and peri-natal conditions and birth outcomes: (1) Pre-pregnancy BMI, calculated from measured height at the first prenatal visit and patient report of prepregnancy weight and categorized according to accepted definitions (obesity ≥ 30 kg/m^2^, overweight 25–29.99 kg/m^2^, healthy weight 18.5–24.99 kg/m^2^, and underweight <18.5 kg/m^2^); (2) Gestational age at birth. All participants received early pregnancy dating by last menstrual period (LMP) and/or early ultrasound, given enrollment criteria. Gestational age at birth was determined from the delivery record using the best obstetrical estimate determined by comparing the date of delivery in relation to the estimated date of confinement established by the 8–14 week prenatal visit considering last menstrual period and/or ultrasound according to standard clinical criteria [38]. Gestational age at birth was classified as preterm (<37 weeks gestation), early term (between 37–0/7 and 38–6/7 weeks), or full term (≥39 weeks); (3) *Cardiometabolic complications of pregnancy*. Women were considered to have gestational hypertension if they had new onset hypertension (defined as systolic blood pressure ≥ 140 mm Hg and/or diastolic blood pressure ≥ 90 mm Hg) at ≥20 weeks gestation in the absence of proteinuria or new signs of end-organ dysfunction (with blood pressure readings documented on at least two occasions at least four hours apart), and they were considered to have preeclampsia if they met the same blood pressure criteria along with proteinuria or new signs of end-organ dysfunction [39]. Women were considered to have gestational diabetes if they developed glucose intolerance after 20 weeks gestation, as diagnosed by an abnormal oral glucose tolerance test [40].

Venous blood draw was completed by a phlebotomist during the study visits. Serum aliquots from the venous blood were obtained and stored at −80 degrees until later analysis of serum lipid profiles.

### 2.3. Chemicals and Lipid Standards

Chemical reagents and HPLC grade organic solvents were purchased from Sigma Aldrich (St. Louis, MO, USA). These include HPLC grade Methanol, Chloroform, Acetonitrile, Isopropanol, Butylated Hydroxytoluene, Ammonium Formate, and Formic Acid. Synthetic lipid standards 21:0–22:6 PC was purchased from Avanti Polar Lipids (Alabaster, AL, USA). Oxylipin standards 9,10-DIHOME, 20-HETE, 9-HETE, 8,9-DHET, 14,15-DHET, 5-HETE, 12-HETE, 11,12-DHET, 5,6-DHET, 13-KODE, 13-HpODE, 9-HpODE, 9-HODE, 9-HoTre, 13-HoTre, 14(15)-EET, 8(9)-EET, 11(12)-EET, 12(13)-EPOME, 13-HODE, Prostaglandin E2, Prostaglandin E2 Ethanolamide, Prostaglandin E2 Glycerol, ARA-Ethanolamide (AEA), 2-AG, Oxy-AEA, Oleamide, Docosanoyl Ethanolamide, Docosahexaneoyl Ethanolamide (DHEA), Linoleoyl Ethanolamide (LEA), dihomo-gamma Linolenoyl Ethanolamide, a-Linolenoyl Ethanolamide (ALEA), Resolvin D3, Resolvin D1, Resolvin D2, Resolvin E1, Thromboxane B2, Leukotriene B4, Prostaglandin F2a, 8-Isoprostane, Oleoyl Ethanolamide (OEA), Palmitoyl Ethanolamide (PEA), Stearoyl Ethanolamide (SEA) were purchased from Cayman Chemical (Ann Arbor, MI, USA). 

### 2.4. Workflow

The targeted lipidomics workflow used for analysis is depicted in Figure 1. Serum samples from Visit 1 and Visit 2 were registered and extracted. The extracted lipids were then analyzed via LC-MS/MS and data was processed. Downstream statistical analysis was conducted to identify differentially abundant lipids.

### 2.5. Lipid Extraction

Patient serum samples were extracted according to a modified Bligh and Dyer lipid extraction protocol [41]. Briefly, 200 μL sample was homogenized in 1 mL methanol:chloroform (2:1) and 0.1% butylated hydroxytoluene (BHT) to prevent the auto-oxidation of polyunsaturated lipid species. Samples were vortexed for 30 min and subsequently centrifuged for 10 min at 4000 rpm to pellet any precipitated protein. The supernatant was collected and phases were separated by the addition of 500 μL 100 mM sodium chloride solution. The organic phase was retained, dried under nitrogen gas, and the weight of recovered lipids was recorded. The extracted lipids were reconstituted in 4 mL methanol:chloroform (1:1) prior to injection into the mass spectrometer. 

### 2.6. Oxylipin and Endocannabinoid Extraction

Oxilipins and endocannabinoids were isolated from serum samples using an automated C18 solid phase extraction manifold (Biotage, Uppsala, Sweden). For this, 100 μL of patient samples was transferred to a deep well 96 well plate. To the sample, 300 μL of 20:80 methanol:water, 55 μL 1% BHT, and 80 μL of glacial acetic acid was added to adjust the pH to 3.0. Samples are centrifuged at 4000 rpm for 10 min at 4 °C to pellet any precipitated solids. The supernatant was transferred to a preconditioned C18 solid phase extraction (SPE) plate (Isolute C18, Biotage, Uppsala, Sweden). The sample was then rinsed with 800 μL water, followed by 800 μL hexane. Oxylipins and endocannabinoids were then eluted with 400 μL methyl formate, dried under nitrogen gas, and subsequently reconstituted in 200 μL methanol for LC/MS analysis.

### 2.7. Chromatography

To quantify DHA containing phospholipids, 20 μL of extracted lipids were injected onto a Thermo Acclaim C30 column (100 × 4.6, Thermo, Waltham, MA, USA) and lipid classes were separated by a 12 minute linear gradient using HPLC Grade water:acetonitrile (Solvent A, 40:60) and isopropanol:acetonitrile (Solvent B, 90:10), each with 0.1 mM Ammonium Formate and 0.1% Formic Acid added as solvent modifiers. To quantify oxylipins and endocannabinoids in patient serum samples, a total of 11 μL oxy/endo extract (10 μL for oxylipin analysis and 1 μL for endocannabinoid analysis, run separately in negative and positive ionization modes, respectively) was deposited onto an Accucore C18 column (100 × 4.6, Thermo, Waltham, MA, USA) and resolved on a 16 minute linear gradient using water as Solvent A and acetonitrile as Solvent B, both containing 10 mM ammonium acetate. For each experiment, the analytical column was heated to 50 °C in a temperature-controlled column chamber and 0.5 mL/min flow rate was used. All relevant gradient information used for chromatographic resolution is provided in Appendix A.

### 2.8. Mass Spectrometry

An Exion AC HPLC system (Sciex, Framingham, MA, USA) with binary pump and autosampler coupled to a QTrap 5500 (Sciex, Framingham, MA, USA) mass spectrometer was used to complete the lipidomics analyses. For identification of DHA containing phospholipids, the mass spectrometer was operated in the negative ion mode. Precursor ion scanning for *m*/*z* 327.4, corresponding to the molecular weight of DHA, was conducted to selectively target the subset of phospholipids that contain esterified DHA (Figure 2). The mass range of *m*/*z* 100–1000 was used for analysis and each of the abundant ions in this range was fragmented for characterization of species.

To detect oxylipins and endocannabinoids, the mass spectrometer was operated in the negative and positive ion modes, respectively. For each, a multiple reaction monitored (MRM) based method was used for detection of species. The table of identified species and their respective MRM transitions are reported in Appendix A.

All instrumental parameters such as electrospray (ESI) voltage, ion spray gases (GS1 and GS2), declustering potential, and collision energy (CE) were optimized using synthetic standards and held consistent over the course of analysis. All instrumental parameters are listed in Appendix A.

### 2.9. Quantification

To quantify DHA containing phospholipids, an external calibration curve in the concentration range of 0.23 μM–4.23 μM was prepared and the equation of the linear trendline was used for quantification. Similarly, for oxylipin and endocannabinoid quantification, the area under the curve of each identified lipid is calibrated against an external calibration curve in the linear range of 0.1 nM–10 nM. The quantified lipids are then used to create a lipid profile and for statistical analysis.

### 2.10. Data Analysis

Identification of DHA containing lipids was done in using LipidView version 1.2 (Sciex, Framingham, MA, USA). Raw mass spectrometry data was uploaded into the database and peak areas were extrapolated and aligned. Parent and fragment ion data was searched against an extensive lipid database, containing over 25,000 lipid species in more than 50 lipid classes, to make putative lipid identifications. All ions with a signal-to-noise ratio above 5 were selected and, as an additional quality filter, only MS/MS confirmed lipid species are reported. All putatively identified lipids were manually confirmed by visual inspection of extracted data. Aligned peak areas were exported and processed using Excel, whereby a table of positively identified lipids, corresponding peak areas, and corrected concentrations was created. Statistical analysis of the resulting table was completed using open source statistics platform, MetaboAnalyst [42]. This analytical tool has recently been used to determine the key metabolic differences in breast cancer models, comparing AA and Caucasian women [43].

## 3. Results

### Analysis

We began our analysis by first comparing patient metadata, such as age, body mass index (BMI), and gestational age of birth, of *n* = 27 African American women with cardiometabolic complications of pregnancy to *n* = 27 AA women who delivered at term and did not have such complications; the latter is used as the control group for this study. Relevant patient metadata is provided in Table 1. There were no statistically significant differences in the age or initial body mass index of women with cardiometabolic complications of pregnancy compared to controls (Figure 3a,b). Moreover, all of the women who participated in the study are at peak maternal age [44].

We next investigated the abundance of serum DHA among study participants (Figure 4). We summed the concentration of all DHA containing phospholipids to calculate total DHA. Using a student’s *t*-test, we identified statistically significant increases in the overall abundance of DHA in serum of both control and CMD patients, comparing Visit 1 to Visit 2 (Figure 4a). However, at Visit 1, there is no statistically significant difference in the abundance of total DHA between CMD and control patients (Figure 4a). When the abundance of DHA is normalized by the weight of total lipid recovered from extracted serum samples, there are significant increases in relative DHA in the serum of CMD patients at Visit 2 by ANOVA (Figure 4b).

We then identified differentially abundant lipids in serum of CMD and control patients at Visit 1 and Visit 2 using one-way ANOVA with Tukey’s HSD post hoc analysis. There are significant increases in DHA containing phospholipid, 36:6 PC (Figure 5). This corresponds to 14:0–22:6 PC.

Figure 6 shows the distribution of lipids at Visit 1. A volcano plot, plotting the log transformations of fold change versus the p-value, demonstrates that concentrations of Resolvin D1 (RvD1), Resolvin E1 (RvE1), 2-arachidonoylglycerol (2-AG), and prostaglandin E2-glyceryl ester (PGE2-Gly) are significantly increased in the serum of control patients compared to CMD patients. Each of these lipids have fold changes (FC) of greater than 1.3 and *p*-values < 0.05.

A volcano plot of lipid distribution in later pregnancy (Visit 2) demonstrates that the abundance of endocannabinoid 2-AG is significantly decreased in the serum of CMD patients compared to control patients (Figure 7). At Visit 2, the abundance of RvE1 is no longer statistically significant (*p*-value 0.069) though also appears to be decreased in CMD serum compared to control patients.

We then sought to evaluate the use of differentially abundant lipids as biomarkers of cardiometabolic diseases of pregnancy by plotting the Receiver Operating Characteristic (ROC) and used the area under the curve (AUC) as the predictive measure (Figure 8). Figure 8 shows that 2-AG, RvD1, and RvE1 have AUC > 0.7, suggesting that these lipids are predictive of cardiometabolic complications of pregnancy in early pregnancy. At Visit 2, RvE1 and 2-AG have an AUC of 0.72 and 0.66, respectively, however, only 2-AG is statistically significant (Figure 8). We then created a multivariate ROC curve using all dysregulated lipids, RvE1, 2-AG, RvD1, PGE2-Gly, and 36:6 PC (Figure 9). Combining the ROC curves of each of the selected lipids increases AUC to 0.759. The average accuracy of the prediction based on 100 cross validations is 73.7%.

A longitudinal analysis of the patient’s serum samples, comparing the lipidome of each patient at Visit 1 and Visit 2, is shown in Figure 10. Primary components analysis (PCA) of the CMD and control groups show that there are moderate changes in the lipidome between Visits 1 and Visits 2 (Figure 10a). For the control group, there are statistically significant decreases in the abundance of 40:8 PS and 36:6 PS (*p*-value < 0.05) (Figure 10b). The abundance of 22:6 LPC has a *p*-value of 0.07 (Figure 10b). A similar pattern is seen in the serum of CMD patients. There are significant decreases in 40:8 PS and 36:6 PC and a significant increase in 22:6 LPC (*p*-value < 0.05).

## 4. Discussion

Docosahexaenoic acid (DHA) is an omega-3, long chain polyunsaturated fatty acid (LCPUFA) with 22 acyl carbons and 6 non-conjugated double bonds. It is basally synthesized from essential fatty acid α-linolenic acid along the Sprecher pathway [45]. DHA is often esterified to membrane phospholipids, predominantly at the sn-2 position of the phosphoglycerol backbone. Once hydrolyzed by phospholipaseA2, DHA can then be acted upon by a series of epoxygenases to synthesize a series of downstream bioactive lipid mediators that have been shown to resolve inflammation. These anti-inflammatory lipids are synthesized along several inflammatory pathways, including cyclooxygenase 2 (COX2) and 5-lipoxygenase (5-LO) pathways [46]. Oxylipins derived from other dietary lipids, such as α-linolenic acid (ALA), linoleic acid (LA), as well as arachidonic acid (ARA) and eicosapentaenoic acid (EPA) are also formed along these same pathways.

The link between DHA and pregnancy outcomes is well established. Generally, it is accepted that increasing plasma levels of DHA is beneficial to pregnancy and is correlated with positive birth outcomes [11,12,13]. However, many of the women who participated in those studies were white, non-Hispanic women and it is unknown if they had underlying heart disease or genetic predispositions. A recent publication reviewing 22 studies relating DHA status to maternal health concluded that it is difficult to draw clear conclusions due to variability in sociodemographics [12]. In this well controlled study, the variability in sociodemographic factors and genome is minimized because all study participants self-identified as Black, non-Hispanic, originating from the United States and have similar economic and education status (Table 1).

For this study, AA women with gestational diabetes, gestational hypertension, or preeclampsia were grouped together for the cardiometabolic complications group. While each of these complications contributes to cardiovascular risk, the etiology varies. Gestational diabetes stems from dysregulation of insulin and altered fatty acid metabolism while hypertension and preeclampsia may arise from underlying kidney disease or changes in the renin–angiotensin or sympathetic nervous systems [19,47,48]. Obesity is a risk factor for all cardiometabolic complications of pregnancy. Our data suggests that the specific type of cardiometabolic risk does not independently affect the distribution of DHA species because all at-risk groups cluster similarly (Appendix A).

The participants of our study experience increases in DHA-containing phospholipids over the course of pregnancy, comparing Visit 1 to Visit 2 (Figure 4a). However, in early pregnancy there were no differences in the total DHA abundance when comparing serum of CMD patients at Visit 1 to control patients at Visit 1. Similarly, at Visit 1 there are no statistically significant changes in relative DHA in CMD patients at Visit 1 and control patients at Visit 1. This suggests that solely monitoring DHA abundance at early pregnancy cannot sufficiently predict cardiometabolic outcomes of pregnancy. We therefore need to monitor the abundance of specific DHA containing lipids. Our findings demonstrate that abundance of 36:6 PC is significantly increased in the serum of CMD patients at Visit 2 compared to CMD patients at Visit 1 and Control patients at Visit 1 and Visit 2 (Figure 5). Taken together with the observation that CMD patients at Visit 2 have significantly higher total DHA than CMD patients at Visit 1, as well as control patients at Visit 1 and Visit 2, this suggests that 36:6 PC (14:0–22:6 PC) is the major driver of this observation. The role of this lipid remains unclear though it has previously been reported in the kidneys of obese mice when mapping the effects of obesity on pregnant and non-pregnant mice and humans and further demonstrated that essential fatty acid derived PUFA are processed differently by obese animals during pregnancy [49]. Similarly, myristic acid (C14:0) is either obtained from dietary sources or the result of de novo free fatty synthesis by the fatty acid synthase (FASN) and deletion of FASN alters the lipid profile of milk produced by lactating mice [50].

To identify key lipids that are dysregulated in early pregnancy that are indicative of cardiometabolic diseases of pregnancy, we investigated the distribution of DHA phospholipids, oxylipins, and endocannabinoids in serum of pregnant AA women at Visit 1. Figure 6 shows that concentrations of RvD1, RvE1, 2-AG, and PGE2-Gly are significantly decreased in the serum of women with cardiometabolic disease compared to the control group. RvD1 is the most significantly dysregulated and is more than 2-fold decreased in the serum of CMD patients. This lipid is synthesized from DHA via 2 main pathways: 15/5 lipoxygenase pathways and the COX-2 pathway [51]. It has been reported that these specialized pro-resolving mediators (SPMs) have more anti-inflammatory properties than the DHA precursor [36]. In inflamed, obese, adipose tissue, treatment of RvD1 and RvD2 decreased the abundance of leptin, TNFα, IL-6, and IL-1B [52]. Our finding is consistent with a previous study of Brazilian women where RvD1 abundance was significantly lower in the plasma of pregnant women with preeclampsia compared to control women, though the ethnic origin of the *n* = 60 women in the study is unclear [51,53]. In a related study following 58 preterm births and 115 controls, RvD1 was found to be the most predictive marker of preterm delivery and other eicosanoids resulting from the lipoxygenase pathway showed the strongest association with preterm childbirth [54]. In addition to RvD1, RvE1 concentration was also significantly lower in the serum of CMD patients. This lipid is also derived from COX2 pathway, though EPA serves as its biosynthetic precursor [36]. The role of this SPM is not well understood. However, in mouse models, administration of RvE3 reduced the occurrence of LPS-induced preterm delivery [51,55].

Figure 6 demonstrates that the concentrations of 2-AG and PGE2-Gly are significantly lower in the serum of pregnant AA women with cardiometabolic disease. Endocannabinoid, 2-AG, is a potent agonist of cannabinoid receptors, CB1 and CB2 [56,57,58]. Regulation of the endocannabinoid system is necessary for pregnancy maintenance and timing of labor [56,57]. Aberrant endocannabinoid signaling has been associated with preeclampsia and early pregnancy loss [57]. Interestingly, 2-AG is metabolized by COX-2 to generate PGE2-Gly [59], and this lipid is also significantly lower in the serum of CMD patients. Contrary to prostaglandins, the role of PGE2-Gly in pregnancy is not known. However, this lipid has been found to play a role in pain and immunomodulation in rats [59]. To the authors’ knowledge, this is the first association of this lipid with cardiometabolic outcomes of pregnancy.

The distribution of 2-AG is dysregulated in late pregnancy (Figure 7). Similar to Visit 1, there are significant decreases in serum concentrations of 2-AG at Visit 2. The concentration of RvE1 appears lower in serum of CMD patients compared to controls but is not statistically significant at this time point (*p*-value 0.069). At Visit 2, concentrations of RvD1 and PGE2-Gly are no longer significantly altered in the plasma of CMD patients compared to controls. Lipid metabolism is variable in late pregnancy as the body prepares for delivery [60,61] and we believe that this variability led to larger error bars in sampling and hindered further identification of differentially abundant lipids, even after normalization with total extracted lipid weight to account for fluid retention in late pregnancy.

To evaluate the predictive ability of 2-AG, RvD1, and RvE1, we plotted ROC curves at Visit 1 and Visit 2 (Figure 8). Panel 8 demonstrates that 2-AG and RvD1 are each predictive of cardiometabolic disease between 8–14 weeks of pregnancy (Visit 1), with AUCs of 0.728 and 0.724, respectively. By weeks 24–30 (Visit 2), RvE1 has an AUC of 0.77 (Figure 8). However, the abundance of this lipid is not significantly different in serum of CMD and control patients (Figure 7). To evaluate the combined predictive strength of the aforementioned bioactive lipids at Visit 1, we plotted a multivariate ROC curve. This increased the AUC to 0.759 with a predictive accuracy of 73.5% and we presume that the predictive strength could potentially be further increased with greater sample size. The identification of key lipids that are predictive of cardiometabolic complications of pregnancy in the AA population is a novel finding, and can potentially lead to therapeutics to address disparities within this underserved community.

Combined, we present data demonstrating that pregnant AA women with cardiometabolic complications of pregnancy have increased levels of serum DHA phospholipids, driven by the abundance of 36:6 PC. Despite increased DHA phospholipids, CMD patients have decreased serum abundance of DHA derived pro-resolving lipid mediator, RvD1, as well as RvE1, 2-AG, and PGE2-Gly. This suggests that these patients have either decreased activity of phospholipase A2 isoforms or separate enzymes, e.g., epoxygenases, necessary for biosynthesis of SPMs. Alterations in PLA2 activity have been reported previously in pregnant women with preeclampsia and gestational diabetes, though the specific PLA2 isoform and mechanism requires further investigation [62,63]. We propose that the decrease of circulating pro-resolving lipid mediators leads to the increased inflammation that is hallmark of cardiometabolic complications of pregnancy and can be used as a blood-based biomarker of these disorders as early as the first trimester. 

## Figures and Tables

**Figure 1 nutrients-15-00140-f001:**
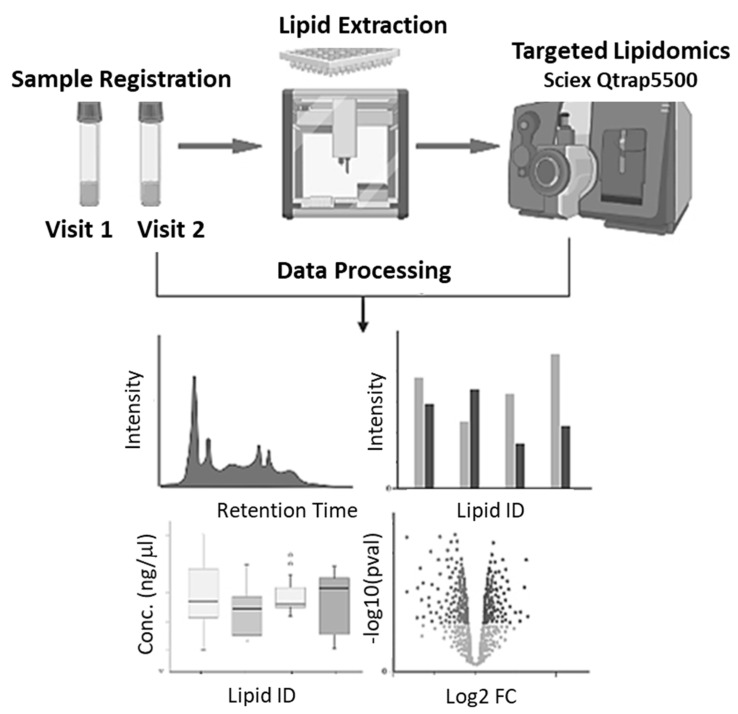
Targeted Lipidomics Workflow. Samples are registered and extracted using an automated sample extraction manifold. Extracted lipids are analyzed by LC/MS in a targeted manner to quantify DHA containing subset of lipids as well as low abundance oxylipins and endocannabinoids. Identified lipids are used to create profiles.

**Figure 2 nutrients-15-00140-f002:**
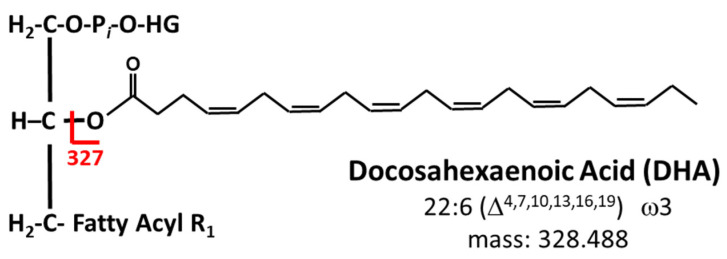
Targeted lipidomics permits the selective targeting of DHA containing phospholipids by conducting precursor ion scans in the negative ion mode for the *m*/*z* value that corresponds to the molecular weight of DHA, *m*/*z* 327.4.

**Figure 3 nutrients-15-00140-f003:**
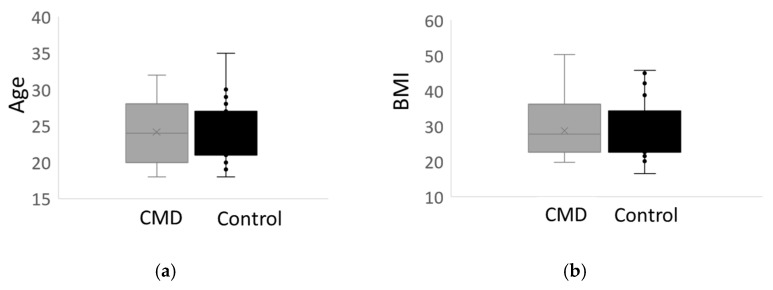
Patient Metadata. (**a**) A whisker plot of the average age of African American women with cardiometabolic disease (CMD) compared to the control group, African American women who delivered at term who do not have cardiometabolic disease. There is no significant difference in the age of the participants. (**b**) Whisker plot comparing the average body mass index (BMI) of the CMD group versus the control group shows that there is no statistical difference in BMI in both cohorts.

**Figure 4 nutrients-15-00140-f004:**
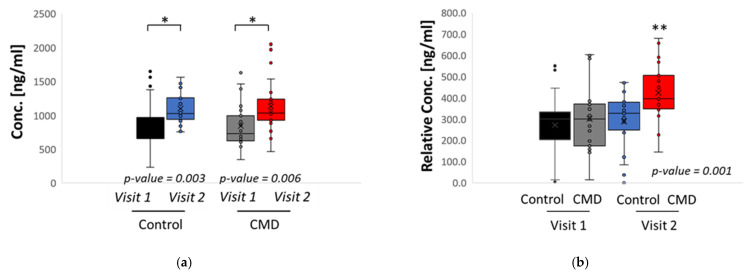
Statistical analysis of patient’s serum. (**a**) Student’s *t*-tests comparing the total concentration of DHA in serum of Control patients at Visit 1 vs. Visit 2 and CMD patients at Visit 1 vs. Visit 2. Each cohort has significant increases in total DHA. (**b**) The concentration of total DHA was normalized by the weight of extracted lipids and an ANOVA and Tukey’s post hoc analysis was conducted to compare both cohorts at Visits 1 and 2. The CMD group has significant increases in relative DHA at Visit 2 and is significantly higher than all other variables. Asterisks indicate statistical significance with * *p* value < 0.05 and ** *p* value < 0.001.

**Figure 5 nutrients-15-00140-f005:**
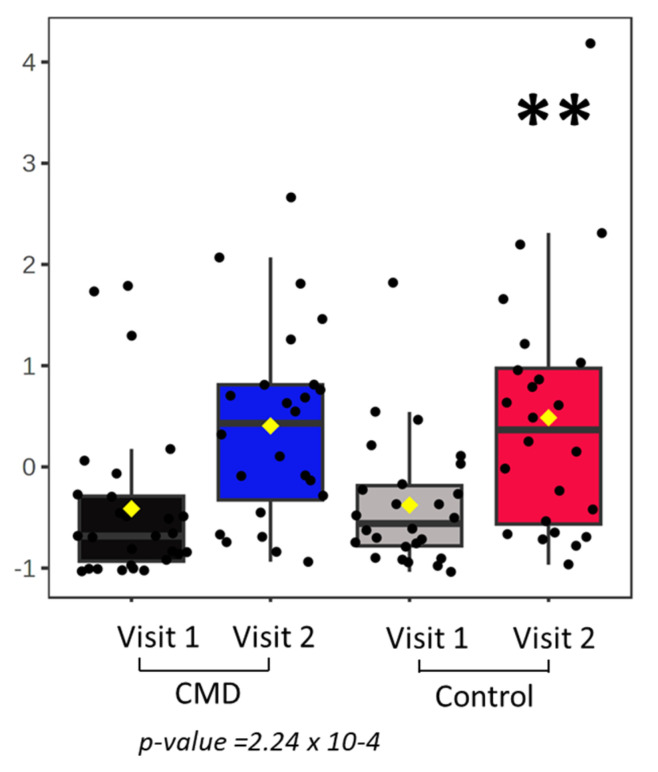
Dysregulation of 36:6 PC. There are significant increases in the distribution 36:6 PC in women with cardiovascular disease compared to the control group by ANOVA with Tukey’s post hoc analysis. The asterisks indicates statistical significance with ** *p* value < 0.001.

**Figure 6 nutrients-15-00140-f006:**
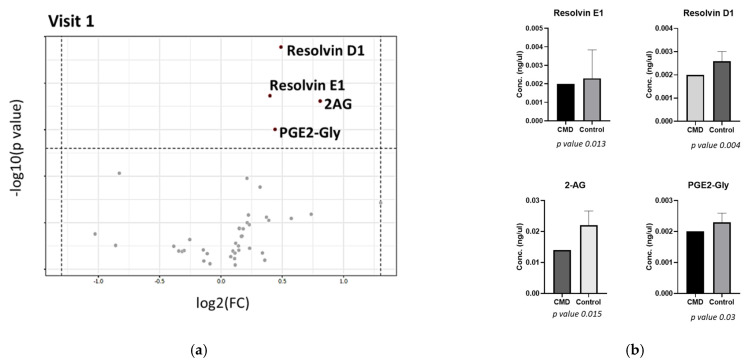
Dysregulation of lipids in early pregnancy. (**a**) Volcano plot showing increased concentrations of Resolvin E1, Resolvin D1, 2-AG, and PGE2-Gly in serum of control patients compared to the CMD group. (**b**) Box plots demonstrating statistically significant increases of RvE1, RvD1, 2-AG, and PGE2-Gly in control group compared to CMD group.

**Figure 7 nutrients-15-00140-f007:**
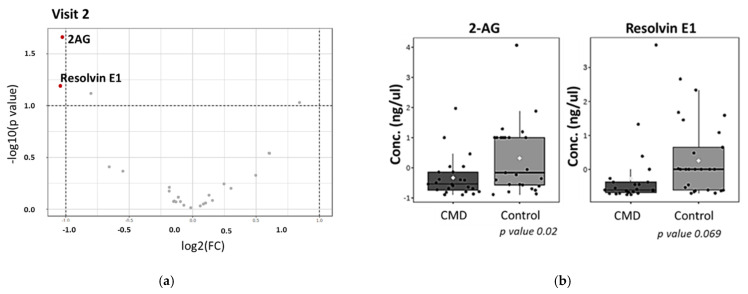
Distribution of oxylipins in late pregnancy. (**a**) Volcano plot demonstrating that there are significant decreases in 2-AG concentrations in the serum of CMD patients compared to control patients. (**b**) There are significant decreases in normalized concentrations of 2-AG in serum of CMD group compared to control group. The abundance of RvE1 appears lower in the serum of CMD patients though is not statistically significant (*p* value 0.069).

**Figure 8 nutrients-15-00140-f008:**
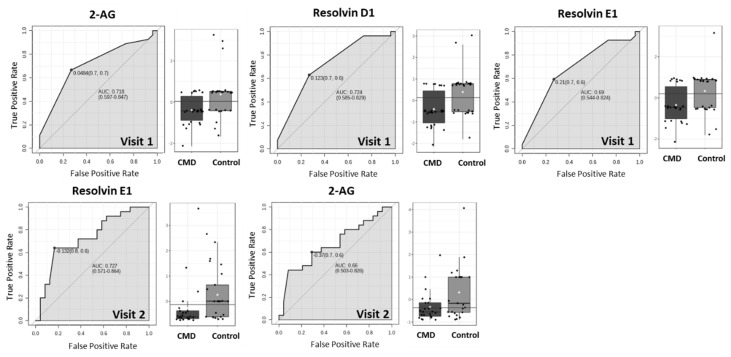
ROC curves to determine predictive strength of dysregulated lipids as biomarkers. At Visit 1, 2-AG, RvD1, and RvE1 have AUC of 0.718, 0.724, and 0.69, respectively. At Visit 2, RvE1 and 2-AG have AUC of 0.727 and 0.66, respectively.

**Figure 9 nutrients-15-00140-f009:**
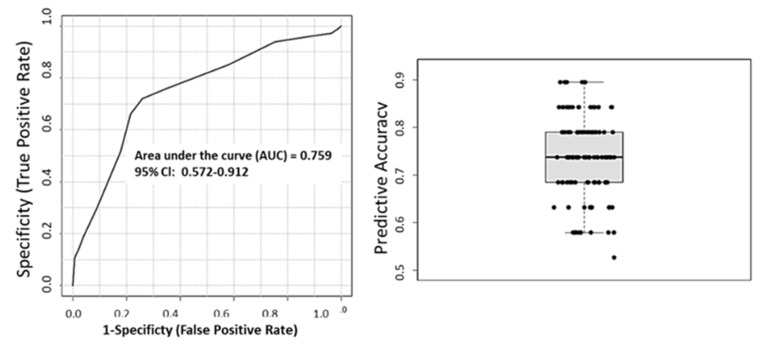
Multivariate ROC curve of differentially abundant lipids. Using a PLS-DA algorithm with 2 latent variables and 5 clusters the AUC is 0.759 and predictive accuracy of 73.7% based on 100 cross validations.

**Figure 10 nutrients-15-00140-f010:**
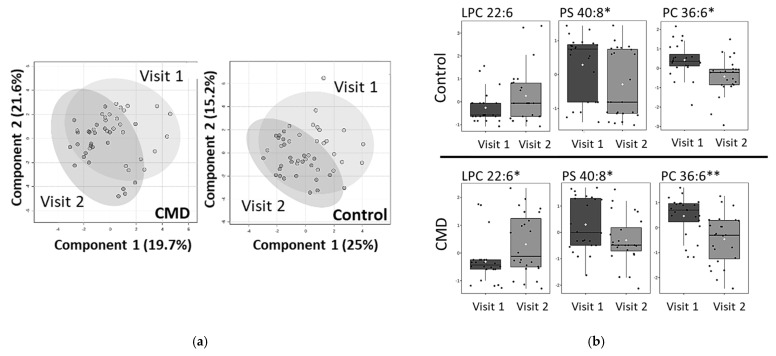
Longitudinal analysis of lipidome (**a**) Principal Component Analysis (PCA) of CMD and Control groups at Visit 1 and Visit 2. (**b**) Whisker plots of statistically significant lipids in control and CMD groups. In control group, 40:8 PS and 36:6 PC have *p* values of 0.04 and 0.002, respectively. LPC 22:6 is not statistically significant (*p* value 0.07). In CMD group, 40:8 PS (*p* value 0.04), 36:6 PC (*p* value 0.001), and 22:6 LPC (*p* value 0.03). Asterisks indicate statistical significance with * *p* value < 0.05 and ** *p* value < 0.001.

**Table 1 nutrients-15-00140-t001:** Characteristics of Study Participants.

	Controls (*n* = 27)	Cases ^a^ (*n* = 27)	*p*-Value ^b^	n
**Sociodemographic Characteristics**				
Maternal age, years	24.0 (3.94)	24.2 (4.66)	0.875	54
**Education Level**			0.792	52
Less than High School Diploma	20	14.8		
High School Diploma	32	37		
Some College	28	25.9		
College Graduate or above	20	22.2		
**Pregnancy Insurance Status**			0.318	52
Medicaid	76	63		
Private	24	37		
**Clinical Measures**				
First prenatal BMI ^c^	30.0 (9.00)	29.4 (8.57)	0.801	52
**First prenatal BMI Category**			0.691	52
Underweight	4	3.7		
Normal weight	36	44.4		
Overweight	12	7.4		
Obese	48	44.4		
Nulliparous	60	66.7		54
Gestational Age at Timepoint 1, weeks	11.3 (2.20)	11.0 (2.01)	0.528	52
Gestational Age at Timepoint 2, weeks	26.9 (2.32)	26.1 (2.45)	0.267	49

Results are presented as mean (SD) for continuous variables and n (%) for categorical variables. ^a^ Women were classified as cases if they experienced GDM or HDP. ^b^ Differences between groups were compared using Student’s *t* test for continuous variables and chi-square tests for categorical variables. ^c^ BMI categorization: Normal, 18.5–24.9 kg/m^2^; Overweight, 25–29.9 kg/m^2^; Obese, 30–39.9 kg/m^2^.

## Data Availability

Not applicable.

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
