# Peer review of "African American Women with Cardiometabolic Complications of Pregnancy Have Decreased Serum Abundance of Specialized Pro-Resolving Lipid Mediators and Endocannabinoids"

_nutrients, 2022, doi:10.3390/nu15010140_

Round 1

Reviewer 1 Report

Summary: The manuscript titled “Specialized Pro-resolving Mediators and endocannabinoids are downregulated in the serum of African American women with cardiometabolic complications of pregnancy” is a research article report employing a targeted lipidomic approach to characterize the distribution of DHA-containing phospholipids and SPMs in maternal serum collected in early and late pregnancy to identify critical lipids that are dysregulated during pregnancy in African-American women who develop cardiometabolic complications. As a result of this approach, the authors successfully identified a lipid signature in early pregnancy serum samples of African-American women that are predictive of cardiometabolic pregnancy complications with close to 75% accuracy. This is a novel and exciting finding that has the potential to be used in the early diagnosis of pregnancy complications specific to a particular population (African-American women). 

The manuscript is very well-written and easy to follow. The content is organized in a logical and easy-to-read manner. The authors provide appropriate well-balanced background information. The methods and experimental design are well-described and proper for the conclusions. Each section of the manuscript is well-defined and contains only relevant information. The figures included in the document are of excellent quality and easy to read and understand. A minor suggestion, if possible, is to make the figures in color, not just black, white and gray.

Author Response

Thank you for your time and thoughtful review of this manuscript. We agree that the figures would be more easily understood with the addition of color. We have added color to Figures 4, 5, as well as Supplementary Figure 1.

Reviewer 2 Report

case-control study cannot be done prospectively. its done only retrospectively.

pregnant women were invited to participate in this on-going cohort study.

its a prospective study . so it is an intervention study

or you need to change the study design

Author Response

Thank you for your comments.

In response to your comments, we have changed the wording of the methods section to reflect that patients were recruited into an on-going study.

The final comment is not clear to me as we have not tried to intervene with the study, simply to better understand the progression of cardiometabolic disease by identifying lipid markers.

Reviewer 3 Report

In their original research article, Kristal M. Maner-Smith and colleagues explore the alterations in the profile of circulating lipids of pregnant African American women with cardiometabolic complications. Using a lipidomic approach, the Authors are successful at showing that the levels of some key lipids are altered. Prior to publishing however, the Authors should address several concerns in order to improve the readability, impact and consistency of their work.

Broad comments

11)     Two important concerns about the title, that in my opinion must be changed:

a)      The target population of the study are African American women, thus, in my opinion, the title is biased by stating “Specialized Pro-resolving Mediators and endocannabinoids are downregulated in the serum of African American women with cardiometabolic complications of pregnancy”, since no matter what the alteration observed it would always be in African American women. The way it is written it seems that the decrease of Specialized Pro-resolving Mediators and endocannabinoids is an exclusive feature of African American women going through cardiometabolic complications during pregnancy. However, it is not the case. In fact, in the discussion the Authors give examples of other studies showing the dysregulation of some of the same lipid species in women with preeclampsia or women that went through preterm childbirth.

b)     In line with comment #13) The use of the term downregulated is too ambitious since no evaluation of RNA or protein levels was performed. This term prompts the reader for a genetic context, which is not the case in the current work.

So, considering these two concerns I would suggest to change the title to something like: “African American women with cardiometabolic complications of pregnancy have decreased levels of specialized pro-resolving mediators and endocannabinoids in the serum.” or if the Authors put together part of the message they deliver in the final paragraph another title can emerge: “Pregnant African American women with cardiometabolic complications of pregnancy have decreased serum abundance of docosahexaenoic acid-derived pro-resolving lipid mediators.”

22)     The Authors use a large number of abbreviations for which they do not indicate what they stand for. Some examples include IL-1, TNFα, PPARγ, TNFR-I and VEGF. Even though most of these abbreviations are used broadly in scientific articles and texts books the Authors should mentioned what they stand for the first time each abbreviation appears in the text and once an abbreviation is defined there is no need to repeat ahead.

33)     Line 77: “pro-resolving SPMs” – Since SPM stands for specialized pro-resolving mediators, pro-resolving SPM is redundant. The Authors should remove pro-resolving.

44)     Line 80-81: “predictive power of the identified lipids” – the Authors should mention the predictive power of predicting what, though the reader might have an idea of what it is considering the information given in the introduction.

55)     Lines 260-263: “When the abundance of DHA is normalized by the weight of total lipid recovered from extracted serum samples, there are significant increases in relative DHA in the serum of CMD patients at Visit 2 by ANOVA (Figure 4b)” – Y axis in figure 4b is ng/ml, however according to “the abundance of DHA is normalized by the weight of total lipid recovered” I was expecting it to be ng/µg ou ng/mg.

Legend of figure 4 does not provide such information, but if the Authors are indeed showing DHA levels normalized by the weight of total lipid recovered it should be stated also in the figure legend.

66)     Please mention in the legend of figure 5 which post hoc test was used.

77)     Some sentences along the manuscript are confusing and hard to understand.

This is an example: “Each of these lipids have fold changes of greater than FC 1.3 and p-values <0.05.” Plus, the Authors do clarify what FC stands for.

Another example is: “Whisker plots of statistically significant lipids in control and CMD groups” in line 319. It is not the lipids that are statistically different, the differences in their levels between visit 1 and visit 2 are. Please rephrase all the sentences where it is stated that the lipids are statistically different.

One more example is “treatment of RvD1 and RvD2” in line 384. It should be read “treatment with RvD1 and RvD2” instead of “treatment of RvD1 and RvD2”.

88)     The Authors state that “2-AG is significantly decreased in the serum of CMD patients compared to control patients” (Lines 286-287). However, in figure 7b, left plot, the value of control group is lower.

99)     Line 297 and 299: “Figure 8a” and “Figure 8b” – Panels in figure 8 are not identified as “a”, “b”, “c”…

110)  Line 297-298: “Figure 8a shows that 2-AG, RvD1, and RvE1 have AUC >0.7, suggesting strong predictive power at Visit 1.” – I do not agree that AUC of 0.718 and 0.724 are high enough to constitute a strong predictive power of cardiometabolic diseases. This also applies to the

111)  Figure 9 is missing.

112)  Legend Figure 10: “LPC 22:6 is approaching significance (p value 0.07).” Either there is a statistical difference in the levels between visit 1 and visit 2 or there is not. “Approaching significance” is not scientifically accurate. The same applies to “trending towards significance (p-value 0.069)” in line 410.

113)  Line 378: “Figure 6 shows that RvD1, RvE1, 2-AG, and PGE2-Gly are significantly downregulated in” – I strongly advise against using the term downregulated. It is commonly used when referring to gene expression and it might be misguided. In fact, the Authors did not measure the levels of RNA or proteins involved in those synthetic pathways. It is preferable to use something like “decreased” or “decreased levels”.

Specific comments

11)     Table 1: Is it n = 27 for cases or is it n = 28 as mentioned in line 99?

22)     Legend of table 1: What HDP stands for?

33)     Line 128: “(defined as systolic blood…” – The parenthesis is not closed.

44)     Line 166: “Briefly, 200 l sample” – Is it 200 µl? Same on line 170.

55)     Line 237: “This analytical tool recently been used to” – “has” is missing: This analytical tool has recently been used to

66)     Line 247: “age of initial body mass index” – “or” instead “of”: age or initial body mass index

Author Response

Responded to comments in the order posed for ease of response.

Broad comments

1)  Two important concerns about the title, that in my opinion must be changed:

  1. a) The target population of the study are African American women, thus, in my opinion, the title is biased by stating “Specialized Pro-resolving Mediators and endocannabinoids are downregulated in the serum of African American women with cardiometabolic complications of pregnancy”, since no matter what the alteration observed it would always be in African American women. The way it is written it seems that the decrease of Specialized Pro-resolving Mediators and endocannabinoids is an exclusive feature of African American women going through cardiometabolic complications during pregnancy. However, it is not the case. In fact, in the discussion the Authors give examples of other studies showing the dysregulation of some of the same lipid species in women with preeclampsia or women that went through preterm childbirth.

  1. b) In line with comment #13) The use of the term downregulated is too ambitious since no evaluation of RNA or protein levels was performed. This term prompts the reader for a genetic context, which is not the case in the current work.

So, considering these two concerns I would suggest to change the title to something like: “African American women with cardiometabolic complications of pregnancy have decreased levels of specialized pro-resolving mediators and endocannabinoids in the serum.” or if the Authors put together part of the message they deliver in the final paragraph another title can emerge: “Pregnant African American women with cardiometabolic complications of pregnancy have decreased serum abundance of docosahexaenoic acid-derived pro-resolving lipid mediators.”

Response: Thank you for your time and thoughtful review of this manuscript. The observations outlined here are very helpful. We have reworded the title to reflect these suggestions. It is now “African American women with cardiometabolic complications of pregnancy have decreased serum abundance of specialized pro-resolving lipid mediators and endocannabinoids”.

  2)  The Authors use a large number of abbreviations for which they do not indicate what they stand for. Some examples include IL-1, TNFα, PPARγ, TNFR-I and VEGF. Even though most of these abbreviations are used broadly in scientific articles and texts books the Authors should mentioned what they stand for the first time each abbreviation appears in the text and once an abbreviation is defined there is no need to repeat ahead.

      Response:  We have written out the name of these cytokines.

3)     Line 77: “pro-resolving SPMs” – Since SPM stands for specialized pro-resolving mediators, pro-resolving SPM is redundant. The Authors should remove pro-resolving.

      Response: Agreed. We have removed pro-resolving before use of SPM.

4)     Line 80-81: “predictive power of the identified lipids” – the Authors should mention the predictive power of predicting what, though the reader might have an idea of what it is considering the information given in the introduction.

      Response: We have edited these lines to indicate that the serum abundance of the identified lipids is predictive of cardiometabolic complications of pregnancy.

5)     Lines 260-263: “When the abundance of DHA is normalized by the weight of total lipid recovered from extracted serum samples, there are significant increases in relative DHA in the serum of CMD patients at Visit 2 by ANOVA (Figure 4b)” – Y axis in figure 4b is ng/ml, however according to “the abundance of DHA is normalized by the weight of total lipid recovered” I was expecting it to be ng/µg ou ng/mg.

Legend of figure 4 does not provide such information, but if the Authors are indeed showing DHA levels normalized by the weight of total lipid recovered it should be stated also in the figure legend.

Response: The total concentration of DHA was normalized by the weight of total lipid recovered and again by the volume of serum used. The figure legend was edited to provide more explanation.

6)     Please mention in the legend of figure 5 which post hoc test was used.

      Response: We have edited the legend to indicate that Tukey’s post hoc test was used.

7)     Some sentences along the manuscript are confusing and hard to understand.

This is an example: “Each of these lipids have fold changes of greater than FC 1.3 and p-values <0.05.” Plus, the Authors do clarify what FC stands for.

Another example is: “Whisker plots of statistically significant lipids in control and CMD groups” in line 319. It is not the lipids that are statistically different, the differences in their levels between visit 1 and visit 2 are. Please rephrase all the sentences where it is stated that the lipids are statistically different.

One more example is “treatment of RvD1 and RvD2” in line 384. It should be read “treatment with RvD1 and RvD2” instead of “treatment of RvD1 and RvD2”.

Response:  We will make it clear that the serum abundance of these lipids is statistically significant.

8)     The Authors state that “2-AG is significantly decreased in the serum of CMD patients compared to control patients” (Lines 286-287). However, in figure 7b, left plot, the value of control group is lower.

      Response:  Thank you for this observation. Though it appears that the concentration of 2-AG is lower in control patients, the median value is higher than the median value of CMD patients. I have changed the type of graphic so that this can be seen more clearly.

9)     Line 297 and 299: “Figure 8a” and “Figure 8b” – Panels in figure 8 are not identified as “a”, “b”, “c”…

      Response:  This has been corrected.

10)  Line 297-298: “Figure 8a shows that 2-AG, RvD1, and RvE1 have AUC >0.7, suggesting strong predictive power at Visit 1.” – I do not agree that AUC of 0.718 and 0.724 are high enough to constitute a strong predictive power of cardiometabolic diseases. This also applies to the

      Response:  The remainder of this comment was missing from the reviewer’s responses. However, in response to what is provided, while I agree that AUC of 0.9 would have been preferable, I respectfully disagree. An AUC of  >0.7 is a respectable measure of predictive power, particularly given our sample size. However, in light of your comment, we have added the caveat that the predictive strength could potentially be increased with greater sample size.

11)  Figure 9 is missing.

      Response:  Figure 9 is shown on page 11 and is provided here for your consideration.

12)  Legend Figure 10: “LPC 22:6 is approaching significance (p value 0.07).” Either there is a statistical difference in the levels between visit 1 and visit 2 or there is not. “Approaching significance” is not scientifically accurate. The same applies to “trending towards significance (p-value 0.069)” in line 410.

      Response: Despite these being widely used terms in scientific publications, we have clarified that the indicated lipids are not statistically significant where indicated.

13)  Line 378: “Figure 6 shows that RvD1, RvE1, 2-AG, and PGE2-Gly are significantly downregulated in” – I strongly advise against using the term downregulated. It is commonly used when referring to gene expression and it might be misguided. In fact, the Authors did not measure the levels of RNA or proteins involved in those synthetic pathways. It is preferable to use something like “decreased” or “decreased levels”.

      Response: This term downregulated has been changed to decreased throughout.

Specific comments

  • Table 1: Is it n = 27 for cases or is it n = 28 as mentioned in line 99? There are n=27 cases. Corrected in text.

2)     Legend of table 1: What HDP stands for? Hypertensive disorders of pregnancy (HDP)

3)     Line 128: “(defined as systolic blood…” – The parenthesis is not closed. Parenthesis added.

4)     Line 166: “Briefly, 200 l sample” – Is it 200 µl? Same on line 170. Corrected to ul

5)     Line 237: “This analytical tool recently been used to” – “has” is missing: This analytical tool has recently been used to. Omitted word has been added.

6)     Line 247: “age of initial body mass index” – “or” instead “of”: age or initial body mass index. Corrected
